# Cell Adhesion Molecule 1 Contributes to Cell Survival in Crowded Epithelial Monolayers

**DOI:** 10.3390/ijms21114123

**Published:** 2020-06-09

**Authors:** Man Hagiyama, Ryuichiro Kimura, Azusa Yoneshige, Takao Inoue, Tomoyuki Otani, Akihiko Ito

**Affiliations:** Department of Pathology, Kindai University Faculty of Medicine, 377-2 Ohno-higashi, Osaka-sayama, Osaka 589-8511, Japan; hagiyama@med.kindai.ac.jp (M.H.); rkimura@med.kindai.ac.jp (R.K.); azusa618@med.kindai.ac.jp (A.Y.); takao@med.kindai.ac.jp (T.I.); otani.tomoyuki@med.kindai.ac.jp (T.O.)

**Keywords:** adhesion molecule, CADM1, epithelial cell, lateral membrane, apoptosis, neutralizing antibody, electroporation

## Abstract

When epithelial cells in vivo are stimulated to proliferate, they crowd and often grow in height. These processes are likely to implicate dynamic interactions among lateral membranous proteins, such as cell adhesion molecule 1 (CADM1), an immunoglobulin superfamily member. Pulmonary epithelial cell lines that express CADM1, named NCI-H441 and RLE-6TN, were grown to become overconfluent in the polarized 2D culture system, and were examined for the expression of CADM1. Western analyses showed that the CADM1 expression levels increased gradually up to 3 times in a cell density-dependent manner. Confocal microscopic observations revealed dense immunostaining for CADM1 on the lateral membrane. In the overconfluent monolayers, CADM1 knockdown was achieved by two methods using *CADM1*-targeting siRNA and an anti-CADM1 neutralizing antibody. Antibody treatment experiments were also done on 6 other epithelial cell lines expressing CADM1. The CADM1 expression levels were reduced roughly by half, in association with cell height decrease by half in 3 lines. TUNEL assays revealed that the CADM1 knockdown increased the proportion of TUNEL-positive apoptotic cells approximately 10 folds. Increased expression of CADM1 appeared to contribute to cell survival in crowded epithelial monolayers.

## 1. Introduction

Glandular and pulmonary epithelial cells in vivo form monolayers of polarized cells with apical and basolateral surfaces [1]. Epithelial cells have the third surface, which consists of the lateral membrane and is important for maintaining the cell polarity [1]. There are particular molecules on the lateral membrane which mediate inter-epithelial cell adhesion [2,3]. Under some physiological conditions, epithelial cells become highly crowded with keeping their monolayer alignment [4]. The most notable examples are columnar epithelial cells in the developing lung and kidney [5,6], endometrial glandular epithelial cells in the proliferative phase [7], and intestinal glandular cells providing a protective barrier against foreign pathogens [8]. These epithelial cells are supposed to continuously receive increased mechanical force on the lateral membrane, which in turn evokes growth-suppressive and/or apoptotic signals within the cells through activating the contact inhibition mechanism involving the Hippo pathway [9,10]. It is speculated that the lateral membrane should be required to change dynamically to adapt the cell to these increased cell–cell contact environments and that the molecules on the lateral membrane should play certain roles in the cell crowding. This speculation, however, has not been examined intensively. There are a number of past studies on epithelial crowding and overcrowding, but they have mainly focused on mechanical aspects of cell–cell contacts and molecular mechanisms for how to balance cell numbers to maintain epithelial homeostasis [4,11,12]. To our knowledge, there are few reports that identified survival factors for crowded epithelial cells.

We have used the polarized 2D cell culture to reproduce epithelial cell morphology in vitro [13,14]. In this culture, epithelial cells are seeded into a culture insert bottomed with a porous semipermeable membrane placed in a multi-well plate. As cells grow in a monolayer, the lateral membrane develops well in some cell lines. As one of the inter-epithelial cell adhesion molecules present on the lateral membrane [2], we have been examining cell adhesion molecule 1 (CADM1), a member of the immunoglobulin superfamily [15,16,17]. CADM1 is expressed not ubiquitously but rather specifically in the limited kinds of epithelial cells, such as bronchiolo-alveolar, renal tubular, endometrial glandular, and gastric fundic glandular cells [18,19,20]. It is noticeable that these cells form simple columnar epithelia in vivo. CADM1 is known to be involved in epithelial cell structure [21]. Therefore, this adhesion molecule is suggested to play a particular role in epithelial cell crowding.

In the present study, we collected a variety of epithelial cell lines that were derived from the simple columnar epithelia that normally express CADM1. We examined how the CADM1 expression would change as the cells crowded in the polarized 2D culture. We also examined whether CADM1 downregulation would affect the cell morphology and cell survival in crowded epithelial cell monolayers.

## 2. Results

### 2.1. CADM1 Expression Increases as Epithelial Cells Crowd

In our past studies, we used NCI-H441 and RLE-6TN cells as pulmonary epithelial cells that expressed CADM1 [22,23]. In this study, we plated these cells on a semipermeable membrane in a two-chamber plate so as to obtain a 30% or 50% confluent cell culture on the next day, and continued the culture for 2 or 2.5 days after the cells reached 90% confluence to allow the cells to become overcrowded (110% confluence). The cells were harvested at various degrees of confluence and were subjected to Western blot analyses for CADM1 (Figure 1A). The CADM1 expression levels increased gradually as the cell confluence increased, and finally reached two to three times those at 50% confluence (Figure 1B).

NCI-H441 cell cultures at 50%, 70%, and 110% confluence were subjected to immunofluorescence analyses assisted by confocal microscopy. As the confluence increased, the immunofluorescent signals for CADM1 became stronger on the lateral membrane, and the cells grew in height, reaching 5.36 µm, the distance between the basal and apical membranes in the Z-stack sectional cell view by confocal microscopy (Figure 1C).

### 2.2. CADM1 Knockdown Induces Apoptosis in Crowded Epithelial Cells

We attempted to knockdown *CADM1* in 110% confluent cell cultures using liposome-based and virus-mediated conventional transfection methods but failed. Then, we devised a pair of electroporation electrodes, which were circle stainless steel plates and placed in the upper and lower chambers to sandwich the semipermeable membrane at a distance of 4 mm (Figure 2). After multiple trials to adjust current-voltage settings, we found the condition where *CADM1*-targeting siRNA reduced the CADM1 level by one-third in NCI-H441 cells at 110% confluence (Figure 3A,B). The same electroporation condition reduced the CADM1 level by half in overcrowded RLE-6TN cells (Figure 3A,B).

CADM1 expression and apoptosis were assessed by immunofluorescence and TUNEL staining, respectively. In control scrambled siRNA-transfected NCI-H441 cell cultures, a clear membranous staining for CADM1 was detected on the lateral membrane between the cells 5.83 µm in height, and TUNEL-positive cells were rarely detected (Figure 3C). In contrast, in *CADM1*-targeting siRNA-transfected cell cultures, membranous staining for CADM1 was weak and often absent from the cell boundary, the cell height roughly halved, and overall, 5% of cells were TUNEL-positive (Figure 3C).

### 2.3. 9D2 Decreases CADM1 Levels and Induces Apoptosis in Crowded Epithelial Cells

The 9D2 anti-CADM1 antibody recognizes the CADM1 ectodomain and blocks *trans*-homophilic binding of CADM1 [24,25,26]. Because 9D2 was able to inhibit cellular process extension in the in vitro hepatocytic ductule formation model [27], we supposed that it might alter the membranous expression of CADM1. We added either 9D2 or control IgY U04 to the upper chamber of NCI-H441 cell cultures at 100% confluence. The cultures were continued for 2 days and were then subjected to Western blot and morphological analyses. We obtained the results similar to those of siRNA experiments. 9D2 reduced the CADM1 expression level by two-thirds, halved the cell height, and made 5% of cells TUNEL-positive (Figure 4 and Figure 5). Similar experiments were conducted on RLE-6TN cells. 9D2 halved the CADM1 expression and made 3% of cells TUNEL-positive (Figure 4 and Figure 5).

We collected six other cell lines of virally transformed or neoplastic epithelial cells that met the two criteria; (1) they express easily detectable levels of CADM1, and (2) they are derived from the normal tissues that expressed CADM1 originally, such as pulmonary, endometrial, and renal tubular epithelia. 9D2 treatment experiments were conducted on these cell lines. In all the cell lines, 9D2 significantly reduced the CADM1 levels by approximately 80% to 50% and caused the emergence of TUNEL-positive cells at increased rates (approximately 3% to 5%) (Figure 4 and Figure 5). 9D2 also decreased the cell height in NCI-H522 and HEC-1-B cells as well as in NCI-H441 cells (Figure 5). The remaining 5 cell lines were too flat (smaller than 1 µm) to be assessed for the changes in height. In the HEC-1-B and OMC-2 cell monolayers, TUNEL-positive cells were often detected in the X-Y planes at or immediately above the monolayer apical surface (for HEC-1-B, compare the two X-Y planes of Figure 5 and Appendix A; for OMC-2, see Appendix A).

We performed reverse-transcription (RT)-PCR experiments to examine whether 9D2 treatment altered the *CADM1* mRNA levels in NCI-H441, NCI-H522, and HEC-1-B cells. There were no differences between U04 and 9D2 treatments in all the three cell lines (Appendix A).

## 3. Discussion

In the present study, we found that the CADM1 expression levels increased as the cells crowded, and that some cell lines grew in heights, and CADM1 was detected clearly on the lateral membrane. We downregulated the increased CADM1 by two methods, siRNA-assisted *CADM1* gene knockdown and neutralizing antibody-assisted CADM1 function blocking, and obtained the consistent results showing that CADM1 downregulation resulted in increased apoptosis in the crowded epithelial cell monolayers. We previously downregulated *cadm1* using siRNA in CNT cells that were grown to 70–80% confluence in a standard culture dish [19]. The reduction in the CADM1 protein level was similar to that by 9D2 in the present study, and apoptosis increased significantly. But, the rate of increase was below 3 folds, and the significance of the difference was just marginal [19]. CADM1 knockdown appeared to induce apoptosis more strongly when epithelial cells are crowded and polarized. Although the precise mechanism by which 9D2 decreases the CADM1 expression remains obscure, the 9D2 treatment did not change the mRNA level for *CADM1* (Appendix A). Therefore, it can be speculated that when 9D2 has interfered with *trans*-homophilic binding of the CADM1 ectodomains between neighboring cells, the CADM1 molecules may become “free” on the cell membrane and then be internalized into the cell to be degraded, as postulated for some cell surface proteins [28].

Originally, we generated the 9D2 neutralizing antibody by immunizing a chicken with a recombinant protein of the mouse CADM1 ectodomain [24]. Our past studies indicate that 9D2 can recognize human and rat CADM1 as well as mouse one [25,27]. Its epitope is unknown, but is supposed to locate within or near the first Ig-like loop of the CADM1 ectodomain, because this loop is structurally the V type, and nectin, another group of the immunoglobulin superfamily structurally closely related to CADM1, is shown to use this loop to bind in trans [2,29]. As we showed here, 9D2 was functional fairly evenly in all cell lines used, including a rabbit one. This may be based on the fact that the CADM1 ectodomain is highly conserved across mammalian species (Appendix A).

What mechanisms link CADM1 downregulation and apoptosis? When epithelial cells crowd in a monolayer, not only apoptotic but also live cells extrude to maintain homeostatic cell numbers, and extruded live cells die from anoikis, a type of apoptosis resulting from cell detachment [4]. Along this scenario, CADM1 downregulation may weaken cell–cell adhesion and promote cell extrusion in the crowded cell monolayers. This simple explanation seems to fit well to the HEC-1-B and OMC-2 cell monolayers, because apoptotic cells were often detected as extruded cells when CADM1 was downregulated. In the other cell monolayers, however, apoptosis was detected in the monolayer composed of crowded cells, suggesting that the apoptotic process proceeds in a cell before the cell extrusion becomes morphologically evident. The Hippo signaling pathway may be among important mechanisms triggering the apoptosis. We previously showed that modest static pressure flattened epithelial cells and could inhibit their growth and slow the cell cycle through activation of YAP, a key transcriptional cofactor of the Hippo pathway [14]. In fact, CADM1 is shown to participate in the control of contact inhibition through the Hippo pathway in NIH3T3 fibroblastic cells and to form complexes with Hippo pathway core molecules LATS1, LATS2, MST1, and MST2 in HEK293 cells [30]. Clinically, co-expression of CADM1 and LATS2 on the cell membrane of cancer cells predicts good prognosis of lung adenocarcinoma [30]. Collectively, CADM1 downregulation may promote apoptosis through involving the Hippo pathway based on the molecular interactions immediately beneath the cell membrane. More general mechanisms may be able to be proposed because actin cytoskeleton is one of the upstream regulators of Hippo signaling capable of activating YAP [31], and actin cytoskeletal derangement initiates apoptosis in epithelial cells [32]. When CADM1 was downregulated in the crowded monolayers, the cell height halved in all the three cell lines whose heights were measurable. CADM1 downregulation is supposed to cause the cell height halving, because CADM1 interacts with cytoskeletons via its intracytoplasmic domain and contributes to maintain epithelial cell morphology [33,34,35]. CADM1 is likely to have similar roles in the other cell lines whose heights were unmeasurable. Epithelial cells may be more susceptible to apoptosis when they crowd because crowding-induced increased mechanical force loaded on the cell can cause cell shape change and cytoskeletal derangement. Crowding epithelial cells may increase CADM1 expression to avoid activating this apoptotic pathway.

In conclusion, CADM1 appeared to increase in various epithelial cell lines as the cells become crowded in monolayers and contribute to cell survival in the overcrowded monolayers. Considering restricted accumulation of CADM1 on the lateral membrane, this function of CADM1 suggests that the lateral membrane integrity and cell–cell adhesion reinforcement may be important for epithelial cell crowding with keeping a monolayer.

## 4. Materials and Methods

### 4.1. Cell Lines

NCI-H441, RLE-6TN, CNT, Caco-2 cells were cultured as described previously [14,19,22,23]. NCI-H522, and 769-P cells were purchased from the American Type Culture Collection (Rockville, MD, USA), and were grown in Roswell Park Memorial Institute medium (RPMI-1640; Wako, Osaka, Japan) supplemented with 10% fetal bovine serum (FBS), antibiotics containing 100 unit/mL penicillin and 100 µg/mL streptomycin (Invitrogen, Carlsbad, CA, USA), and 5 mM HEPES buffer (Dojindo, Kumamoto, Japan) at 37 °C in 5% CO2/95% air. HEC-1-B and OMC-2 were purchased from Riken BioResource Center (Tsukuba, Japan) and Japanese Collection of Research Bioresources (Ibaraki, Japan), respectively, and were grown in Eagle’s minimum essential medium and Ham’s F12 medium, respectively, supplemented with 10% FBS, antibiotics containing 100 unit/mL penicillin and 100 µg/mL streptomycin. All experimentation using these cell lines proceeded within 3 months or 5 passages after resuscitation. Characteristics of the cell lines used are described in detail in Appendix A.

### 4.2. Antibodies

Antibodies against the CADM1 ectodomain (3E1 and 9D2, chicken monoclonal) and the C-terminus (rabbit polyclonal) were described previously [24,36]. 3E1 and 9D2 were used for CADM1 immunofluorescence coupled with TUNEL staining, and CADM1 downregulation in culture, respectively. The C-terminal antibody was used for Western blotting analyses and single-staining immunofluorescence. An anti-β-actin antibody was purchased from Medical & Biological Laboratories (Nagoya, Japan). Peroxidase-conjugated secondary antibodies were purchased from Amersham (Buckinghamshire, England). A chicken IgY clone (U04) was purchased from R&D Systems (Minneapolis, MN, USA).

### 4.3. Cell Culture

In order to start the cell culture at low cell density, 3 × 10^4^ of NCI-H441 and 5 × 10^4^ of RLE-6TN cells were seeded onto the bottom of a culture insert lined with a semipermeable membrane (Transwell, transparent PET membrane, pore size 0.4 µm; Falcon, Corning, Tokyo, Japan) placed in a 12-well plate. NCI-H441 cells reached 30, 50, 70, 90, and 110% confluence after 1, 2, 4, 6, and 8 days, respectively. RLE-6TN cells reached 50, 70, 90, and 110% confluence after 1, 2.5, 5, and 7.5 days, respectively. For siRNA electroporation and 9D2 treatment experiments, 1 × 10^5^ H441, H522, or 769-P cells, 1.5 × 10^5^ HEC-1-B cells, and 2 × 10^5^ RLE-6TN, OMC-2, CNT, or Caco-2 cells were seeded into a Transwell, and reached 100% confluence after 2 days.

### 4.4. Electroporation

A square-wave electroporator CUY21 EDIT (BEX Co., Ltd., Tokyo, Japan) was used with a pair of stainless-steel circle electroporation electrodes that were originally devised by BEX co., Ltd. so as to adjust to the 12-well plate culture system (LF510-8; Figure 2). When NCI-H441 or RLE-6TN cells reached 100% confluence in a Transwell insert after 2 days of cell seeding, cells were washed twice with Opti-MEM I (Fisher Scientific, Leicestershire, UK). A well of a 12-well plate was filled with 1 mL of Opti-MEM I, and the lower electrode was placed on the well bottom. The Transwell insert with cells was placed in the well, and filled with 0.5 mL of Opti-MEM I that contained 1.5 µg of either *CADM1*-targeting siRNA (5′-AACGAAAGACGTGACAGTGAT-3′) or scrambled sequence control RNA described previously [19,37]. The upper electrode was placed in the medium parallel to the Transwell membrane at a distance of 4 mm from the lower electrode. Then, a single electronic pulse was generated at 40 V for 10 msec. Immediately afterwards, the medium was removed gently from the Transwell, and the new one was poured. The cells were cultured for 2 days.

### 4.5. Western Blot Analysis

Cells cultured on a semipermeable membrane were lysed in a buffer containing 50 mM Tris-HCl (pH 8.0), 150 mM NaCl, 1% Triton X-100 and 1 mM phenylmethylsulfonyl fluoride, and after removal of impurities by centrifugation, were subjected to Western blot analyses as described in our previous report [26]. Immunoreactive band intensities were quantified using ImageJ software (National Institutes of Health, Bethesda, MD, USA), as described previously [38].

### 4.6. Immunofluorescence and Confocal Microscopy

Immunofluorescence was performed as previously described [13]. Briefly, NCI-H441 cells were grown on a semipermeable membrane to reach 50, 70, and 110% confluence, fixed in cold methanol, blocked with 2% bovine serum albumin (BSA), and incubated with the anti-CADM1 C-terminal antibody, then visualized with Alexa Flour 488-conjugated secondary antibody (anti-rabbit IgG; Jackson ImmunoResearch, West Grove, PA, USA). Fluorescence images were captured using a C2+ confocal scanning system equipped with 488 nm argon and 543 nm helium–neon lasers (Nikon, Tokyo, Japan). The vertical sectional (Z-plane) images were generated by Z-stack confocal microscopy using a 0.4 µm motor step. Captured images were analyzed on the Nikon C2+ computer system using Analysis Controls tools. Cell heights were measured with Annotations and Measurements at randomly selected 30 cells per Transwell membrane, and their average was calculated. All measurements were performed in triplicate, and the mean and standard error of the cell heights were calculated for each experimental group. The immunofluorescence experiments were repeated three times with similar results.

### 4.7. 9D2 Treatment and Apoptosis Detection

When cells reached 100% confluence in a Transwell insert after 2 days of cell seeding, 9D2 or control IgY (U04) was added to the Transwell medium at a concentration of 10 µg/mL. After 2 days, apoptosis was detected by TUNEL method using the Click-iT™ TUNEL Alexa Fluor™ 594 Imaging Assay (ThermoFisher, Waltham, MA, USA) as described previously [19]. Briefly, cells were washed with phosphate buffered saline (PBS), fixed in 4% paraformaldehyde for 15 min, and treated with 0.25% Triton X-100 in PBS. After washed twice with deionized water, the cells were incubated with the TUNEL reaction mixture containing terminal deoxynucleotidyl transferase 1 h at 37 °C, followed by reaction with Alexa-594-labelled dUTP for 30 min at 37 °C. Soon afterward, CADM1 immunofluorescence was conducted on the TUNEL-stained cells, according to the procedures described previously [26]. Briefly, the cells were incubated with 3E1 anti-CADM1 antibody, then visualized with Alexa Flour 488-conjugated secondary antibody (anti-chicken IgY; Jackson ImmunoResearch), followed by nuclear counterstaining with DAPI (Dojindo, Kumamoto, Japan). Triple fluorescence images were captured using a C2+ confocal scanning system equipped with 488-nm argon and 543 nm helium–neon lasers (Nikon, Tokyo, Japan). The vertical sectional (Z-plane) images were generated by Z-stack confocal microscopy using a 0.4 µm motor step. A cell was deemed TUNEL-positive if it exhibited TUNEL signals among the DAPI nuclear stain. The number of TUNEL-positive cells was counted among 200 cells per Transwell membrane. All measurements were performed in triplicate, and the mean and standard error of the proportion of TUNEL-positive cells were calculated for each experimental group. The TUNEL assays were repeated three times with similar results.

### 4.8. RT-PCR

Total RNA was extracted from the cultured cells at the same time point as the protein extraction for Western blot analyses. Procedures for RNA extraction, RT-PCR, and electrophoresis were described previously [36].

### 4.9. Statistical Analysis

Ratios of Western blot intensities, and cell heights of three groups were analyzed using one-way ANOVA among all experimental groups, and the Bonferroni correction was applied to two particular groups. TUNEL-positive proportions and cell heights of two groups were analyzed using the paired two-tailed Student *t* test. A *p*-value ≤0.05 was considered to indicate statistical significance. Statistical data were shown in detail in Appendix A.

## Figures and Tables

**Figure 1 ijms-21-04123-f001:**
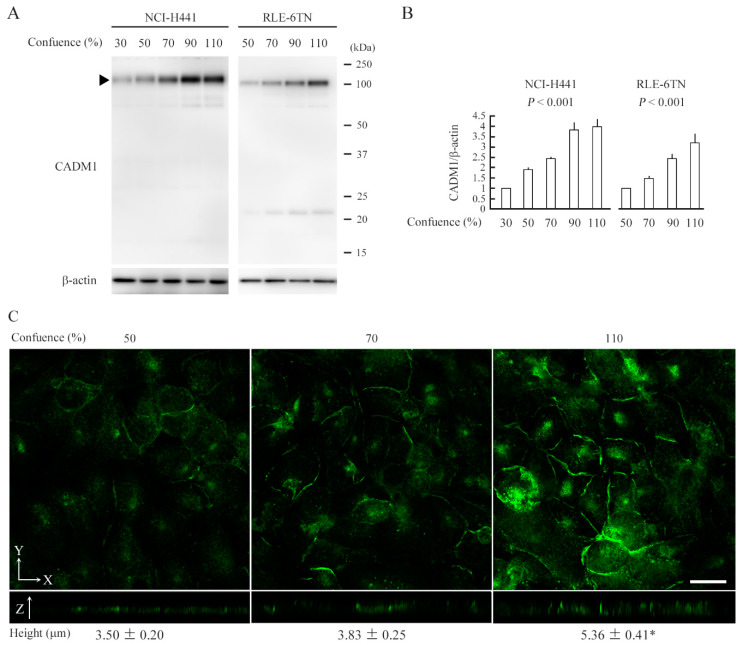
NCI-H441 and RLE-6TN cells express more CADM1 and grow in height as they crowd. NCI-H441 and RLE-6TN cells were cultured on a semipermeable membrane in 12-well plates. The cell lysates were extracted at indicated confluence and were blotted with the antibodies indicated (**A**). Intensities of the immunoreactive bands for CADM1 (the full-length form depicted by an arrowhead) and β-actin were measured densitometrically. The mean ratios of CADM1 to β-actin and their standard deviations were calculated from the data obtained in triplicate experiments. Statistical differences were analyzed by one-way ANOVA, and *p*-values are shown (**B**). NCI-H441 cells at indicated confluence were stained green with CADM1 immunofluorescence using the anti-*C*-terminal antibody. The means and standard deviations of cell heights were calculated from the data obtained in triplicate experiments and were statistically analyzed by one-way ANOVA (**C**). * *p* < 0.001 by Bonferroni correction when compared with 70% confluence cultures (**C**). Scale bar = 10 µm. Statistic data are shown in detail in Appendix A.

**Figure 2 ijms-21-04123-f002:**
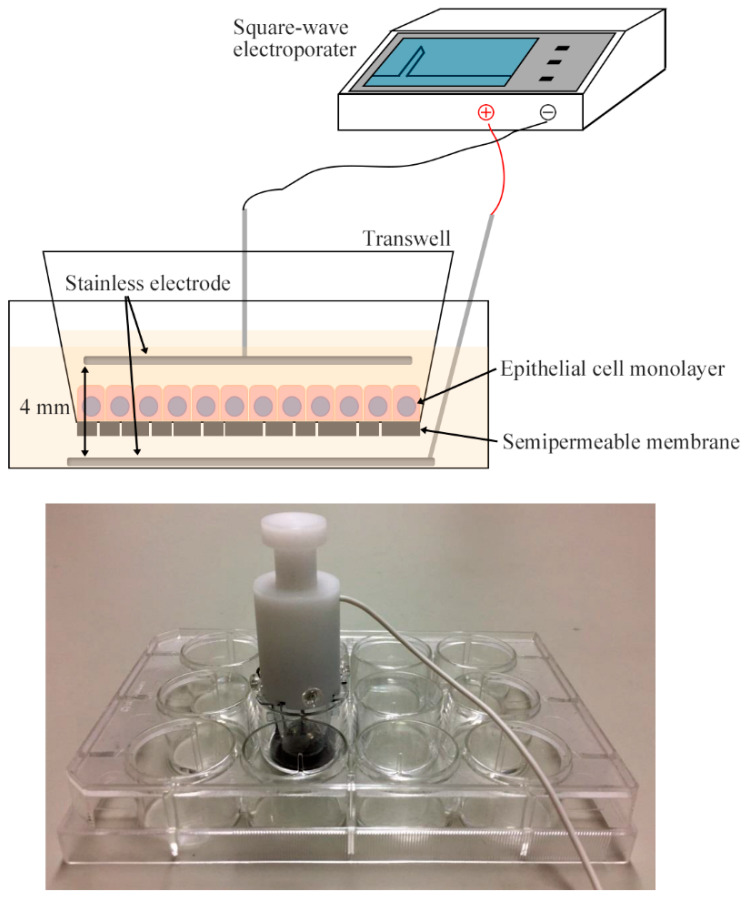
A new electroporation system devised for the polarized 2D cell culture. A pair of stainless-steel circle electrodes were devised so as to adjust to the 12-well plate culture system. The lower electrode was set on the well bottom, and the upper electrode was placed parallel to the Transwell membrane at a distance of 4 mm from the lower electrode. The electrodes were connected with a square-wave electroporator, which was set to generate single pulse at 40 V for 10 msec. Upper, schematic presentation; lower, a photograph of the electrode set in a 12-well plate.

**Figure 3 ijms-21-04123-f003:**
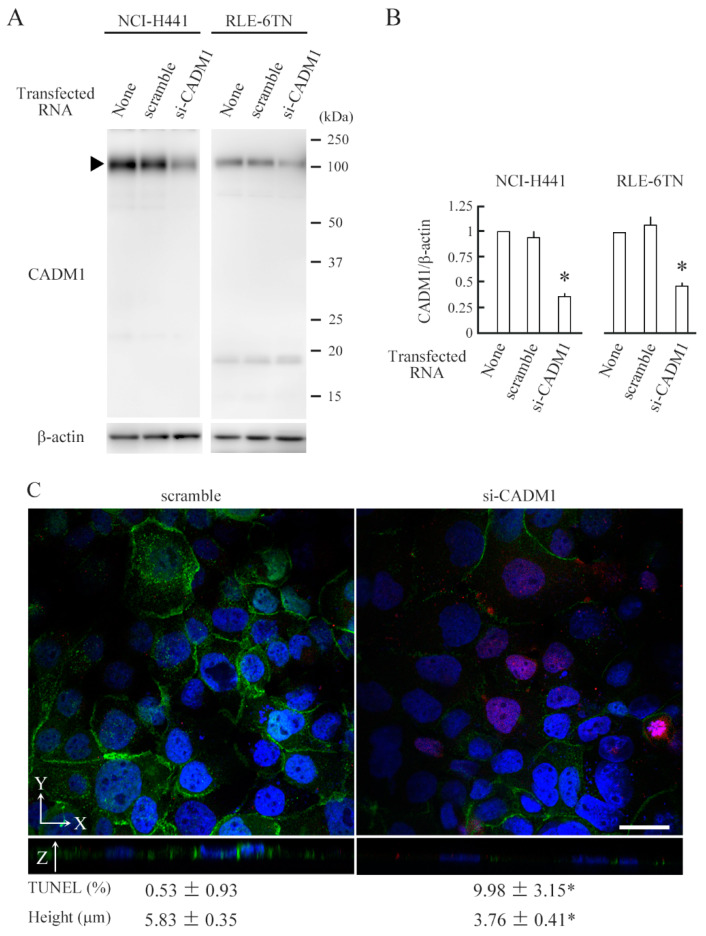
CADM1 downregulation flattens NCI-H441 epithelial cells and induces apoptosis. NCI-H441 and RLE-6TN cells were cultured on a semipermeable membrane in 12-well plates. When the cells reached 100% confluence, they were transfected with scramble RNA or *CADM1*-targeting siRNA (si-CADM1), or without RNA, by electroporation. After 2 days, the cell lysates were extracted, and were blotted with the antibodies indicated (**A**). Intensities of the immunoreactive bands for CADM1 (the full-length form depicted by an arrowhead) and β-actin were measured densitometrically. The mean ratios of CADM1 to β-actin and their standard deviations were calculated from the data obtained in triplicate experiments. Statistical differences were analyzed by one-way ANOVA (see Appendix A). * *p* < 0.001 by Bonferroni correction when compared with scramble RNA transfection. (**B**). After 2 days of transfection, NCI-H441 cells were triple-stained with CADM1 immunofluorescence (3E1 antibody; green), TUNEL method (red), and DAPI nuclear staining (blue). The means and standard deviations of TUNEL-positive cell proportions and cell heights were calculated from the data obtained in triplicate experiments (**C**). * *p* < 0.01 by Student’s *t*-test when compared with scramble RNA transfection. Scale bar = 10 µm.

**Figure 4 ijms-21-04123-f004:**
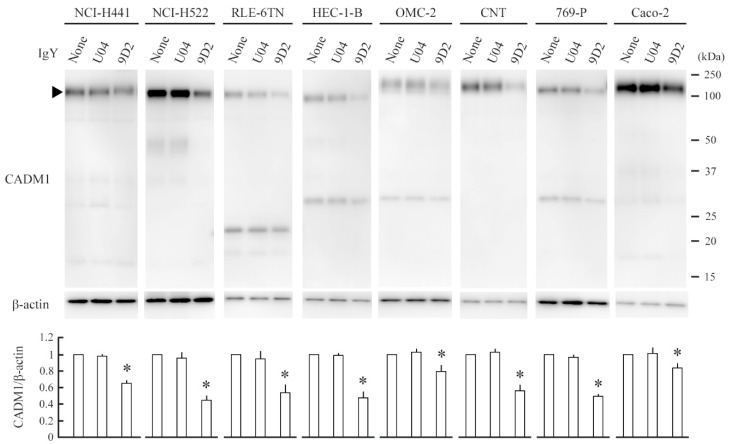
9D2 decreases the CADM1 expression levels in crowded epithelial cells. Various epithelial cell lines were cultured on a semipermeable membrane in 12-well plates. When the cells reached 100% confluence, control IgY U04 or 9D2 was added at a concentration of 10 µg/mL, or no antibody was added (none). After 2 days, the cell lysates were extracted, and were blotted with the antibodies indicated (upper panel). Intensities of the immunoreactive bands for CADM1 and β-actin were measured densitometrically. The mean ratios of CADM1 to β-actin and their standard deviations were calculated from the data obtained in triplicate experiments. Statistical differences were analyzed by one-way ANOVA (see Appendix A). * *p* < 0.01 by Bonferroni correction when compared with the U04 treatment (lower panel).

**Figure 5 ijms-21-04123-f005:**
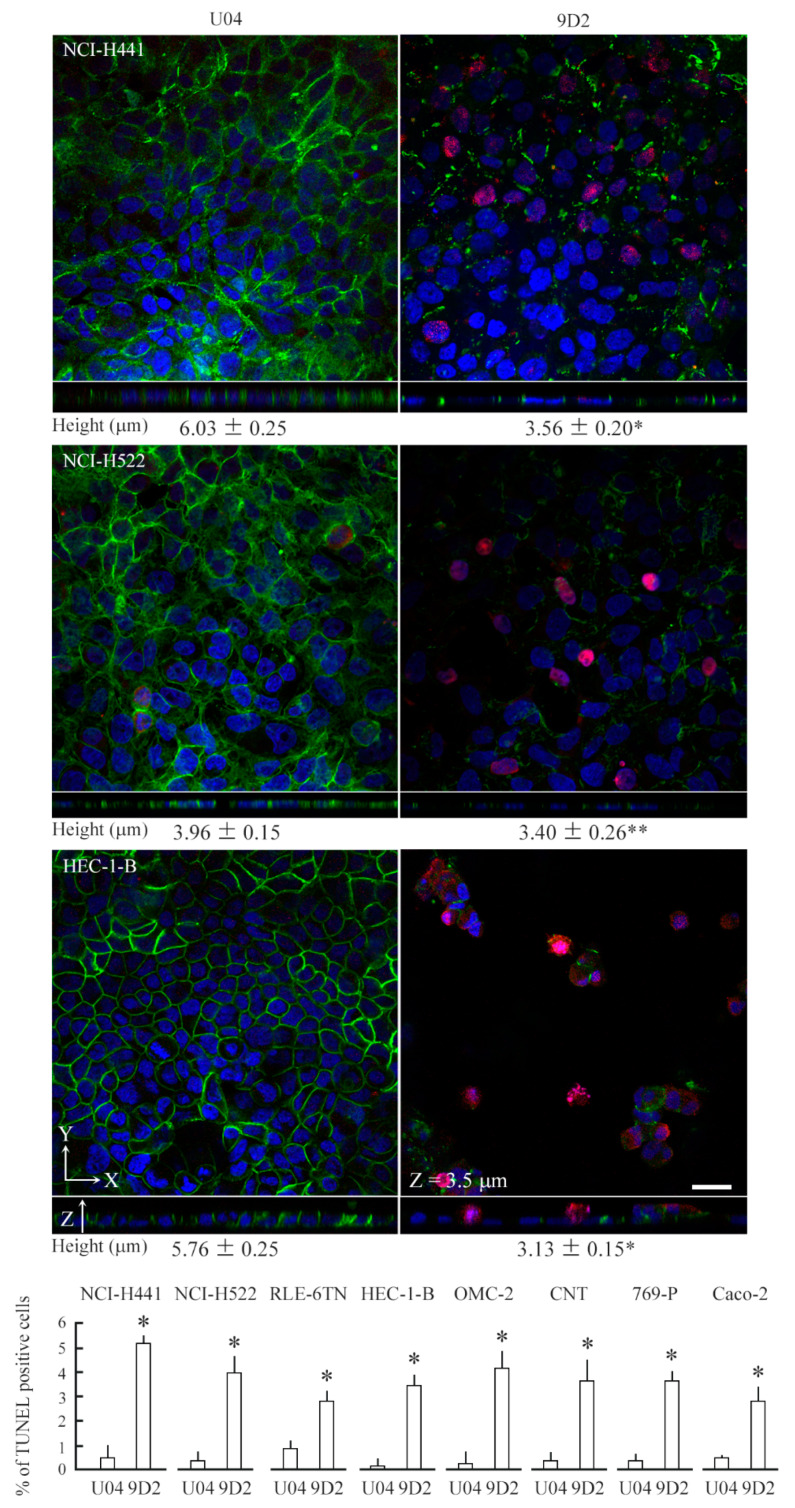
9D2 induces apoptosis in crowded epithelial cells and decreases the cell height. Various epithelial cell lines were cultured on a semipermeable membrane in 12-well plates. When the cells reached 100% confluence, control IgY U04 or 9D2 was added at a concentration of 10 µg/mL. After 2 days, the cells were triple stained with CADM1 immunofluorescence (3E1 antibody; green), TUNEL method (red), and DAPI nuclear staining (blue). The means and standard deviations of TUNEL-positive cell proportions and cell heights were calculated from the data obtained in triplicate experiments. Representative photomicrographs of NCI-H441, NCI-H522, and HEC-1-B cells are shown with the cell height values (upper 3 panels). Note that HEC-1-B cells treated with 9D2 were micrographed in an X-Y plane at the Z axis of about 3.5 µm. TUNEL assay data are shown in the lowest panel. * *p* < 0.01, and ** *p* = 0.03 by Student’s *t*-test when compared with the U04 treatment. Scale bar = 20 µm.

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
