# Peer review of "Cell Adhesion Molecule 1 Contributes to Cell Survival in Crowded Epithelial Monolayers"

_ijms, 2020, doi:10.3390/ijms21114123_

Round 1

Reviewer 1 Report

Hagiyama M. et al. describe the role of cell adhesion molecule 1 (CADM1) in cell survival in overcrowded epithelial monolayers. In particular, they knockdown CADM1 function by two different approaches and show that cells undergo apoptosis.

This work is in line with others from Ito A. concerning epithelia and CADM1 role and introduces an interesting technique for transducing peculiar cell culture.

Generally, the paper is well written especially the technical side (Materials and Methods and Figures), but data are poorly discussed and contextualized, a more in-depth analysis is needed.

The following comments are mean to improve the paper and make it more attractive and comprehensible. 

For authors' convenience, remarks are divided according to the respective sections.

MAJOR POINTS

1- Lines 32-42 represent approximately the only theoretical background provided since the subsequent part of the “introduction” section contain other types of information. Even if clear and well written, this part is too short and does not explain enough the background of the study. Moreover, there is no mention of cancer and CADM1 role in epithelial tumors. Since authors claim that, based on their findings, CADM1 can be considered as a “therapeutic target of early-stage neoplasm” (lines 26 and 213-214) it is mandatory to thoroughly contextualize this study in the tumor setting.

In my opinion, this is the weakest point of this research. It is not clear if authors are interested in describing the molecular role of CADM1 in epithelia from a very basic point of view or if they would provide evidence for considering CADM1 crucial in epithelial cancer. My feeling is that this research is a molecular biology study with not enough data to support a “translational” role for CADM1, thus this kind of speculation seems overstated.

2- Line 112 (and relative main text) claims that “CADM1 downregulation flattens epithelial cells and induces apoptosis” but this is demonstrated only for NCI-H441 since for RLE-6TN only downregulation of CADM1 protein is showed. Without immunofluorescence and TUNEL assay the reader can't confirm this statement. Please, provide the necessary data or adapt this overstatement to the provided data.

3- Lines 127-128: antibody derived from clone 9D2 is used for knockdown experiments and is here introduced. Understandably, authors would like to encourage to read their previous paper, but they must provide in this context the necessary information to enable readers to understand what 9D2 is and how it works. These two lines are not enough.

4- Lines 132-134: this is an overstatement. In none of the panels in Figure 5 TUNEL-positive cells are grouped and do not associate with areas where CADM1 is most downregulated, in fact, at least for NCI-H441 is just the opposite.

5- Line 146 introduces 6 more cell lines, both human and animal, tumoral and not. This inconsistency correlates with the first issue (point 1 of this list), is unclear if authors are working on cancer or in basic molecular biology. This uncertainty undermines the robustness of results and confuses the take-home message. Authors say they choose cell models based on a good CADM1 expression (line 147), but if the main goal is to identify CADM1 as a therapeutic target (lines 26 and 213-214) they must go beyond their previous works (lines 63-64) and concentrate their efforts on tumor models.

6- Lines 147-149: defining all this different cell line as “derived from the normal tissues that expressed CADM1 originally, such as…” is at least simplistic. Tumors do not simply “derive” from an epithelium, tumorigenesis process if far more complex than this. If authors want to focus on cancer, maybe they should remove the non-human model and deepen the analysis of their results based on tumor features. On the other hand, remove all referral to cancer and concentrate generally on epithelial models and CADM1 role in their survival.

7- Line 149: antibody 9D2 is used on cell line both human and non-human but the efficacy of this antibody was assessed in mouse cells (reference 17). Is it reasonable to suppose that there could be interspecies differences in the epitope recognized by 9D2? Can authors discuss this point?

8- Lines 149-151: “In all the cell lines, …caused the 150 emergence of TUNEL-positive cells at increased rates (approximately 3 to 5%)” but data are provided only for 3 cell lines, not for all. The same issue as for Figure 3, see point 2 of this list.

9- Lines 152-153: “The remaining 6 cell lines were too flat (smaller than 1 μm) to be assessed for the changes in height”, this suggests that immunofluorescences were performed in all cell lines, so why did authors do not show them (at least in supplementary data)? Even if cell height is not measurable, CADM1 and TUNEL stainings would be fundamental to sustain authors' statements (see the previous point).

10- Lines 174-178: It’s unfortunate that authors produced and described 9D2 antibody but do not investigate (here or in previous work) its molecular functioning. At least assessing the CADM1 mRNA levels and/or a half-life assay would give a rough view of the molecular process, probably confirming this suggested theory. Besides, authors maybe want to discuss how siRNA and antibody, clearly having two completely different ways to knockdown CADM1 functions, lead to similar phenotype in the same timeframe of 2 days (lines 115, 121, 140, 157).

11- Lines 182-184 introduce Supplementary Figure which is profoundly connected to Figure 5. It is impossible to understand completely HEC-1-B phenotype in Figure 5 without considering Supplementary Figure. Please, remove this from “discussion” and insert it at the end of paragraph 2.3. Moreover, which is the authors’ explanation for all the other cell models? If the subsequent sentence is correct (lines 184-186), maybe more and later time points could clarify this point and support this speculation.

12- Line 196: “The limitation of this study is the limited number of cell lines examined” The major limitation of this study involves cell line, but the problem is not the number. See previous points 1,5 and 6 of this list.

13- Lines 201-202: to my knowledge, there is no information in literature or from producers that these cancer cell models are pre-malignant or early-stage. If authors base this conclusion only on the defined doubling time, this is not enough. A deep analysis of the genomic, transcriptomic and histological profile of a cell line is necessary to reach this conclusion. If there are data demonstrating this, please provide citations. Otherwise, carefully correct this overstatement.

This is again a simplification of the difference between an epithelial tumor and a normal epithelium (point 6 of this list) and is not acceptable.

14- References are appropriate and recent but 13 out of 27 belong to papers from the same research group or, at least, from groups involving the PI of the present work. Maybe authors would enrich the paper with other points of view.

MINOR POINTS

1- Lines 13-24: there are too many details for the “abstract” section, please summarize and/or remove

2- Line 45: authors say that cells were cultured in 6-well plate but everywhere else is stated 12-well plate

3- Lines 45-47: these details are inappropriate for the “introduction” section, please remove

4- Lines 55-60: again, these technical details are not needed in the “introduction” section

5- Lines 85-86: “*” is defined as “P<0,001” but none of the panels “*” are present. Moreover, there is some discrepancy in Fig1B: figure legend report that normalization is considered upon “70% confluence cultures” but in the main text (line 70) it’s reported 50%; panel B clearly show that NCI-H441 are normalized on 30% confluence while RLE-6TN on 50%. Please correct descriptions in legend and main text

6- Lines 119 and 305: Supplementary Table 2 is cited but not provided

7- Line 152: “9D2 also decreased the cell height in NCI-H522 and HEC-1-B cells”, this decrease is reported to be significant also for NCI-H441 considering the “*” reported on the panel. Correct the figure or the text accordingly.

8- Lines 168-169: authors repeat culture condition and again this is not the appropriate section, please remove

9- Line 174: “increased apoptosis in the crowded epithelial cell monolayers” could be an overstatement, please discuss further. As stated in lines 114 and 157, TUNEL experiments were performed only in 100% confluence culture, thus it is not verified if the same effect should be observed even at a lower confluence, where CADM1 expression is lower but clearly present (Figure 1).

10- Line 180: “crowding induces live cell extrusion to maintain homeostatic cell numbers in epithelia in vivo" this is EXACTLY the title of reference 4. Please rephrase this concept.

11- Line 217: “NCI-H441, RLE-6TN, CNT, Caco-2 cells were described previously [9, 14, 18, 19].”, these references are not the paper who first established these cell lines, thus the sentence would be more precise this way “NCI-H441, RLE-6TN, CNT, Caco-2 cells were culture as described previously [9, 14, 18, 19].”

Author Response

Point-by-point Responses to Reviewers of Manuscript ijms-815388

Responses to Reviewer 1

Hagiyama M. et al. describe the role of cell adhesion molecule 1 (CADM1) in cell survival in overcrowded epithelial monolayers. In particular, they knockdown CADM1 function by two different approaches and show that cells undergo apoptosis.

This work is in line with others from Ito A. concerning epithelia and CADM1 role and introduces an interesting technique for transducing peculiar cell culture.

Generally, the paper is well written especially the technical side (Materials and Methods and Figures), but data are poorly discussed and contextualized, a more in-depth analysis is needed.

The following comments are mean to improve the paper and make it more attractive and comprehensible. 

For authors' convenience, remarks are divided according to the respective sections.

Ans) We deeply thank Reviewer for careful and critical reading of our manuscript and valuable comments and indications. Accordingly, we revised our manuscript largely.

MAJOR POINTS

1- Lines 32-42 represent approximately the only theoretical background provided since the subsequent part of the “introduction” section contain other types of information. Even if clear and well written, this part is too short and does not explain enough the background of the study. Moreover, there is no mention of cancer and CADM1 role in epithelial tumors. Since authors claim that, based on their findings, CADM1 can be considered as a “therapeutic target of early-stage neoplasm” (lines 26 and 213-214) it is mandatory to thoroughly contextualize this study in the tumor setting.

In my opinion, this is the weakest point of this research. It is not clear if authors are interested in describing the molecular role of CADM1 in epithelia from a very basic point of view or if they would provide evidence for considering CADM1 crucial in epithelial cancer. My feeling is that this research is a molecular biology study with not enough data to support a “translational” role for CADM1, thus this kind of speculation seems overstated.

Ans) We thank Reviewer for valuable comments and indications, which we understood well. In the revised manuscript, we removed all the referrals to cancer and speculations about CADM1’s roles in cancer biology. Keywords were changed accordingly.

In addition, we added several sentences to the first and second paragraphs of “Introduction” to explain the background of the present study from a basic point of view for epithelial biology (lines 55-56, 57-61, 62-66, and 76-77 in the MS Word format).

2- Line 112 (and relative main text) claims that “CADM1 downregulation flattens epithelial cells and induces apoptosis” but this is demonstrated only for NCI-H441 since for RLE-6TN only downregulation of CADM1 protein is showed. Without immunofluorescence and TUNEL assay the reader can't confirm this statement. Please, provide the necessary data or adapt this overstatement to the provided data.

Ans) Thank you for indicating our overstatement. We inserted “NCI-H441” into this sentence to adapt it to the experimental fact (line 522).

3- Lines 127-128: antibody derived from clone 9D2 is used for knockdown experiments and is here introduced. Understandably, authors would like to encourage to read their previous paper, but they must provide in this context the necessary information to enable readers to understand what 9D2 is and how it works. These two lines are not enough.

Ans) We understood this comment, and inserted some sentences to describe what 9D2 is and how it works (lines 126-128).

4- Lines 132-134: this is an overstatement. In none of the panels in Figure 5 TUNEL-positive cells are grouped and do not associate with areas where CADM1 is most downregulated, in fact, at least for NCI-H441 is just the opposite.

Ans) Thank you for careful reading. We realized that our description was confusing, because it was not based on quantitative analyses. According to the Reviewer’s comment, we deleted this sentence.

5- Line 146 introduces 6 more cell lines, both human and animal, tumoral and not. This inconsistency correlates with the first issue (point 1 of this list), is unclear if authors are working on cancer or in basic molecular biology. This uncertainty undermines the robustness of results and confuses the take-home message. Authors say they choose cell models based on a good CADM1 expression (line 147), but if the main goal is to identify CADM1 as a therapeutic target (lines 26 and 213-214) they must go beyond their previous works (lines 63-64) and concentrate their efforts on tumor models.

Ans) We understand and agree with this Reviewer’s criticism. According to it, we removed all the referrals to cancer and speculations about CADM1’s roles in cancer biology throughout the revised manuscript.

6- Lines 147-149: defining all this different cell line as “derived from the normal tissues that expressed CADM1 originally, such as…” is at least simplistic. Tumors do not simply “derive” from an epithelium, tumorigenesis process if far more complex than this. If authors want to focus on cancer, maybe they should remove the non-human model and deepen the analysis of their results based on tumor features. On the other hand, remove all referral to cancer and concentrate generally on epithelial models and CADM1 role in their survival.

Ans) We understand and agree with this Reviewer’s criticism. According to it, we removed all the referrals to cancer and speculations about CADM1’s roles in cancer biology throughout the revised manuscript.

7- Line 149: antibody 9D2 is used on cell line both human and non-human but the efficacy of this antibody was assessed in mouse cells (reference 17). Is it reasonable to suppose that there could be interspecies differences in the epitope recognized by 9D2? Can authors discuss this point?

Ans) Thank you for letting us realize the necessity of explanations of 9D2, the central player in the present study. Accordingly, we created one paragraph in “Discussion” to provide the basic information supporting that 9D2 could be functional across mammalian species (lines 177-185; Supplementary Figure 4).

8- Lines 149-151: “In all the cell lines, …caused the 150 emergence of TUNEL-positive cells at increased rates (approximately 3 to 5%)” but data are provided only for 3 cell lines, not for all. The same issue as for Figure 3, see point 2 of this list.

Ans) In response to this Reviewer’s comment, we provided the TUNEL staining images for the five cell lines of the rest (Supplementary Figure 2).

9- Lines 152-153: “The remaining 6 cell lines were too flat (smaller than 1 μm) to be assessed for the changes in height”, this suggests that immunofluorescences were performed in all cell lines, so why did authors do not show them (at least in supplementary data)? Even if cell height is not measurable, CADM1 and TUNEL stainings would be fundamental to sustain authors' statements (see the previous point).

Ans) There is a typo. “The remaining 6 cell lines” is not correct. “5 cell lines” is correct, and was revised (line 146). We provided the TUNEL staining images for these five cell lines (Supplementary Figure 2).

10- Lines 174-178: It’s unfortunate that authors produced and described 9D2 antibody but do not investigate (here or in previous work) its molecular functioning. At least assessing the CADM1 mRNA levels and/or a half-life assay would give a rough view of the molecular process, probably confirming this suggested theory. Besides, authors maybe want to discuss how siRNA and antibody, clearly having two completely different ways to knockdown CADM1 functions, lead to similar phenotype in the same timeframe of 2 days (lines 115, 121, 140, 157).

Ans) In response to the Reviewer’s comment, we performed RT-PCR experiments to assess the CADM1 mRNA levels after 9D2 treatment. The method is described in “Materials and Methods” (lines 328-331), and the results are shown in Supplementary Figure 3, and are commented on in “Results” (lines 152-155) and “Discussion” (lines 169-171).

11- Lines 182-184 introduce Supplementary Figure which is profoundly connected to Figure 5. It is impossible to understand completely HEC-1-B phenotype in Figure 5 without considering Supplementary Figure. Please, remove this from “discussion” and insert it at the end of paragraph 2.3. Moreover, which is the authors’ explanation for all the other cell models? If the subsequent sentence is correct (lines 184-186), maybe more and later time points could clarify this point and support this speculation.

Ans) In response to the Reviewer’s comment, we described the HEC-1-B and OMC-2 cell phenotypes with citing Supplementary Figures X and Y at the end of the last paragraph but one in Subsection 2.3.

In addition, we added several sentences at the end of the third paragraph of “Discussion” to present our more general mechanism model for the CADM1–apoptosis linkage (lines 206-218). We considered that the shortening of cell heights should be involved in this linkage.

12- Line 196: “The limitation of this study is the limited number of cell lines examined” The major limitation of this study involves cell line, but the problem is not the number. See previous points 1,5 and 6 of this list.

Ans) We understand this comment is related to the Reviewer’s criticism directed to the uncertain standpoint of our original manuscript. This paragraph was deleted in the revised manuscript.

13- Lines 201-202: to my knowledge, there is no information in literature or from producers that these cancer cell models are pre-malignant or early-stage. If authors base this conclusion only on the defined doubling time, this is not enough. A deep analysis of the genomic, transcriptomic and histological profile of a cell line is necessary to reach this conclusion. If there are data demonstrating this, please provide citations. Otherwise, carefully correct this overstatement.

This is again a simplification of the difference between an epithelial tumor and a normal epithelium (point 6 of this list) and is not acceptable.

Ans) We understand and agree with this Reviewer’s criticism. According to it, we removed all the referrals to cancer and speculations about CADM1’s roles in cancer biology throughout the revised manuscript. We also largely revised the “Introduction” (mainly the first paragraph) and “Discussion” (mainly the first and third paragraphs) sections to clarify the standpoint of the present study and claim the significance in epithelial cell biology.

14- References are appropriate and recent but 13 out of 27 belong to papers from the same research group or, at least, from groups involving the PI of the present work. Maybe authors would enrich the paper with other points of view.

Ans) Thank you for the positive suggestion. In this revision, we added 11 articles independent of us to the reference list (lines between 378 and 485), with which we think our manuscript became much understandable in a straightforward manner.

MINOR POINTS

1- Lines 13-24: there are too many details for the “abstract” section, please summarize and/or remove

Ans) In response to the comment, we removed several sentences.

2- Line 45: authors say that cells were cultured in 6-well plate but everywhere else is stated 12-well plate

Ans) These descriptions correctly reflect our experimental facts. To avoid confusion, we replaced “6-well plate” with “multi-well plate” in this line (line 70).

3- Lines 45-47: these details are inappropriate for the “introduction” section, please remove

Ans) We removed the corresponding sentences.

4- Lines 55-60: again, these technical details are not needed in the “introduction” section

Ans) We removed the corresponding sentences. Instead, we inserted several sentences to indicate the purpose of the present study (lines 82-85).

5- Lines 85-86: “*” is defined as “P<0,001” but none of the panels “*” are present. Moreover, there is some discrepancy in Fig1B: figure legend report that normalization is considered upon “70% confluence cultures” but in the main text (line 70) it’s reported 50%; panel B clearly show that NCI-H441 are normalized on 30% confluence while RLE-6TN on 50%. Please correct descriptions in legend and main text

Ans) “*” is marked on the value (3.56 ± 0.41) at the bottom of panel C. This value indicates the cell height of NCI-H441 cells at 110% confluence. The description about this panel (Figure 1 C) appears in lines 71 to 74 of the original manuscript. On the other hand, the line 70 sentence of the original manuscript describes the CADM1 expression levels based on Western blot analyses (Figure 1 panels A and B). Therefore, we would like to believe there is no discrepancy.

In the revised manuscript, we inserted “(C)” at the end of the “* P < 0.001 by Bonferroni …” sentence in the legend of Figure 1 to indicate where the asterisk is (line 511).

6- Lines 119 and 305: Supplementary Table 2 is cited but not provided

Ans) We are sure we uploaded this table at the first time of submission. This table may have been lost during the editorial conversion process. We confirmed again that this table is attached at the end of the “Supplementary materials” file.

7- Line 152: “9D2 also decreased the cell height in NCI-H522 and HEC-1-B cells”, this decrease is reported to be significant also for NCI-H441 considering the “*” reported on the panel. Correct the figure or the text accordingly.

Ans) For NCI-H441 cells, the line 131-132 sentence says “9D2 reduced the CADM1 expression level by two-thirds, halved the cell height, and …” in the original manuscript. This sentence is rather far from line 152, although these two sentences comment on the same figure panel. For better readability, we added “as well as in NCI-H441 cells” at the end of the line 152 sentence in the revised manuscript (line 145).

8- Lines 168-169: authors repeat culture condition and again this is not the appropriate section, please remove

Ans) We removed the corresponding statement.

9- Line 174: “increased apoptosis in the crowded epithelial cell monolayers” could be an overstatement, please discuss further. As stated in lines 114 and 157, TUNEL experiments were performed only in 100% confluence culture, thus it is not verified if the same effect should be observed even at a lower confluence, where CADM1 expression is lower but clearly present (Figure 1).

Ans) Thank you for indicating the important point to be discussed. In the revised manuscript, we cited our past data of CNT cells, compared them with the present data from the same cell line, and discussed the significance of CADM1 in crowded monolayers (lines 163-169).

10- Line 180: “crowding induces live cell extrusion to maintain homeostatic cell numbers in epithelia in vivo" this is EXACTLY the title of reference 4. Please rephrase this concept.

Ans) We revised this sentence with using anoikis as the keyword (lines 187-194).

11- Line 217: “NCI-H441, RLE-6TN, CNT, Caco-2 cells were described previously [9, 14, 18, 19].”, these references are not the paper who first established these cell lines, thus the sentence would be more precise this way “NCI-H441, RLE-6TN, CNT, Caco-2 cells were culture as described previously [9, 14, 18, 19].”

Ans) Thank you for careful reading. We corrected the sentence accordingly (lines 228-229).

Reviewer 2 Report

The  authors present a compact story on the role of CADM1 in controlling epithelial cell morphology and overcrowding. The experiments have been presented well and western blots have been quantified. Overall the study is rigorous with respect to replicates and cell lines used. Here are few points that could be improved upon.

  1. Please review the title of the paper. Apoptosis was used as a functional assay to shown CADM1's role in maintaining epithelial cell height. Hence the title needs to be redressed to some thing more broad and should not be focused on a assay.
  2. The usage of the word 'tall' seems inappropriate. Epithelial cells are defined by width and height. Hence may be sections can be reworded to address how CADM1 is affecting height of the cells.

Author Response

Point-by-point Responses to Reviewers of Manuscript ijms-815388

Responses to Reviewer 2

The authors present a compact story on the role of CADM1 in controlling epithelial cell morphology and overcrowding. The experiments have been presented well and western blots have been quantified. Overall the study is rigorous with respect to replicates and cell lines used. Here are few points that could be improved upon.

  1. Please review the title of the paper. Apoptosis was used as a functional assay to shown CADM1's role in maintaining epithelial cell height. Hence the title needs to be redressed to some thing more broad and should not be focused on a assay.

Ans) We thank Reviewer for valuable suggestions. Accordingly, we removed “apoptosis” from the title, and instead used “survival”. In addition, “overcrowded” is something specific, and thus, was replaced with “crowded”. The new title is as follows (lines 4-5 in the MS Word format).

Cell adhesion molecule 1 contributes to cell survival in crowded epithelial monolayers

  1. The usage of the word 'tall' seems inappropriate. Epithelial cells are defined by width and height. Hence may be sections can be reworded to address how CADM1 is affecting height of the cells.

Ans) Thank you again for critical reading of our manuscript. In agreement with this comment, we removed “tall” throughout the revised manuscript, and defined the cell height as the distance between the basal and apical membranes in the Z-stack sectional cell view by confocal microscopy (lines 102-104).

Round 2

Reviewer 1 Report

I really appreciate the Authors' efforts. I think that they have addressed all the major concerns and now the manuscript is improved and clearer.